# EDIS: Entity-Driven Image Search over Multimodal Web Content

**Siqi Liu**[1*] **Weixi Feng**[2*] **Tsu-jui Fu**[2] **Wenhu Chen**[3] **William Yang Wang**[2]

[1]Cornell University    [2]UC Santa Barbara
[3]University of Waterloo, Vector Institute
sl2322@cornell.edu,  wenhuchen@uwaterloo.ca
{weixifeng, tsu-juifu, william}@cs.ucsb.edu

## Abstract

Making image retrieval methods practical for real-world search applications requires significant progress in dataset scales, entity comprehension, and multimodal information fusion. In this work, we introduce **E**ntity-**D**riven **I**mage **S**earch (EDIS), a challenging dataset for cross-modal image search in the news domain. EDIS consists of 1 million web images from actual search engine results and curated datasets, with each image paired with a textual description. Unlike datasets that assume a small set of single-modality candidates, EDIS reflects real-world web image search scenarios by including a million multimodal image-text pairs as candidates. EDIS encourages the development of retrieval models that simultaneously address cross-modal information fusion and matching. To achieve accurate ranking results, a model must: 1) understand named entities and events from text queries, 2) ground entities onto images or text descriptions, and 3) effectively fuse textual and visual representations. Our experimental results show that EDIS challenges state-of-the-art methods with dense entities and the large-scale candidate set. The ablation study also proves that fusing textual features with visual features is critical in improving retrieval results. [1]

## 1   Introduction

Image search, also known as text-to-image retrieval, is to retrieve matching images from a candidate set given a text query. Despite the advancements in large-scale vision-and-language models (Wang et al., 2021; Zhang et al., 2021; Chen et al., 2020; Li et al., 2020b), accurately retrieving images from a large web-scale corpus remains a challenging problem. There remain several critical issues: 1) Lack of large-scale datasets: existing image retrieval datasets typically contain 30K-100K images

---

*Equal contribution
[1]Code and data: https://github.com/emerisly/EDIS

Figure 1: EDIS contains entity-rich queries and multimodal candidates. EDIS requires models to recognize subtle differences across different modalities to identify the correct candidates. For instance, the last three sample candidates either miss entities in the image or describe a different event.

(Plummer et al., 2015; Lin et al., 2014), which is far less than the number of images that search engines must deal with in real applications. 2) Insufficient entity-specific content: existing datasets focus on generic objects without specific identities. Specific entities (*"Statue of Liberty"*) in web images and text may be recognized as general objects (*"building"*). 3) Modality mismatch: existing image retrieval methods usually measure image-text similarity. However, for web image search, the surrounding text of an image also plays a crucial part in this fast and robust retrieval process.

Recently, there has been a continuous interest in event-centric tasks and methods in the news domain (Reddy et al., 2021; Varab and Schluter,

2021; Spangher et al., 2022). For instance, NYTimes800K (Tran et al., 2020), and Visual News (Liu et al., 2021) are large-scale entity-aware benchmarks for news image captioning. TARA (Fu et al., 2022) is proposed to address time and location reasoning over news images. NewsStories (Tan et al., 2022) aims at illustrating events from news articles using visual summarization. While many of these tasks require accurate web image search results as a premise, a large-scale image retrieval dataset is lacking to address the challenges of understanding entities and events.

Therefore, to tackle the aforementioned three key challenges, we introduce a large-scale dataset named Entity-Driven Image Search (EDIS) in the news domain. As is shown in Fig.1, EDIS has a much larger candidate set and more entities in the image and text modalities. In addition to images, text segments surrounding an image are another important information source for retrieval. In news articles, headlines efficiently summarize the events and impress readers in the first place (Panthaplackel et al., 2022; Gabriel et al., 2022). Hence, to simulate web image search with multi-modal information, we pair each image with the news headline as a textual summarization of the event. As a result, EDIS requires models to retrieve over image-headline candidates, which is a novel setup over existing datasets.

Given a text query, existing models can only measure query-image or query-text similarity alone. BM25 (Robertson et al., 2009), and DPR (Karpukhin et al., 2020) fail to utilize the visual features, while vision-language models like VisualBert (Li et al., 2019) and Oscar (Li et al., 2020b) cannot be adopted directly for image-headline candidates and are infeasible for large-scale retrieval. Dual-stream encoder designs like CLIP (Radford et al., 2021) are efficient for large-scale retrieval and can compute a weighted sum of query-image and query-text similarities to utilize both modalities. However, as is shown later, such multi-modal fusion is sub-optimal for EDIS. In this work, we evaluate image retrieval models on EDIS and reveal that the information from images and headlines cannot be effectively utilized with score-level fusion. Therefore, we further proposed a feature-level fusion method to utilize information from both images and headlines effectively. Our contribution is three-fold:

- We collect and annotate EDIS for large-scale

image search, which characterizes single-modality queries and multi-modal candidates. EDIS is curated to include images and text segments from open sources that depict a significant amount of entities and events.

- We propose a feature-level fusion method for multi-modal inputs before measuring alignment with query features. We show that images and headlines are exclusively crucial sources for accurate retrieval results yet cannot be solved by naive reranking.

- We evaluate existing approaches on EDIS and demonstrate that EDIS is more challenging than previous datasets due to its large scale and entity-rich characteristics.

## 2 Related Work

**Cross-Modal Retrieval Datasets** Given a query sample, cross-modal retrieval aims to retrieve matching candidates from another modality (Bain et al., 2021; Hu and Lee, 2022; Wang et al., 2022; Sangkloy et al., 2022). Several datasets have been proposed or repurposed for text-to-image retrieval. For instance, MSCOCO (Lin et al., 2014), and Flickr-30K (Plummer et al., 2015) are the two widely used datasets that consist of Flickr images of common objects. CxC (Parekh et al., 2021) extends MSCOCO image-caption pairs with continuous similarity scores for better retrieval evaluation. Changpinyo et al.(2021) repurposes ADE20K (Pont-Tuset et al., 2020) for image retrieval with local narratives and mouse trace. WebQA (Chang et al., 2022) has a similar scale to EDIS and defines source retrieval as a prerequisite step for answering questions. The source candidates are either text snippets or images paired with a short description. In contrast, EDIS is a large-scale entity-rich dataset with multi-modal candidates that aligns better with realistic image search scenarios.

**Text-to-Image Retrieval Methods** Text-to-image retrieval has become a standard task for vision-language (VL) understanding (Lu et al., 2019; Li et al., 2020a; Kim et al., 2021; Jia et al., 2021; Fu et al., 2021; Dou et al., 2022). Single-stream approaches like VisualBert (Li et al., 2019), and UNITER (Chen et al., 2020) rely on a unified transformer with concatenated image and text tokens as input. Dual-stream approaches like CLIP (Radford et al., 2021) or ALIGN (Jia

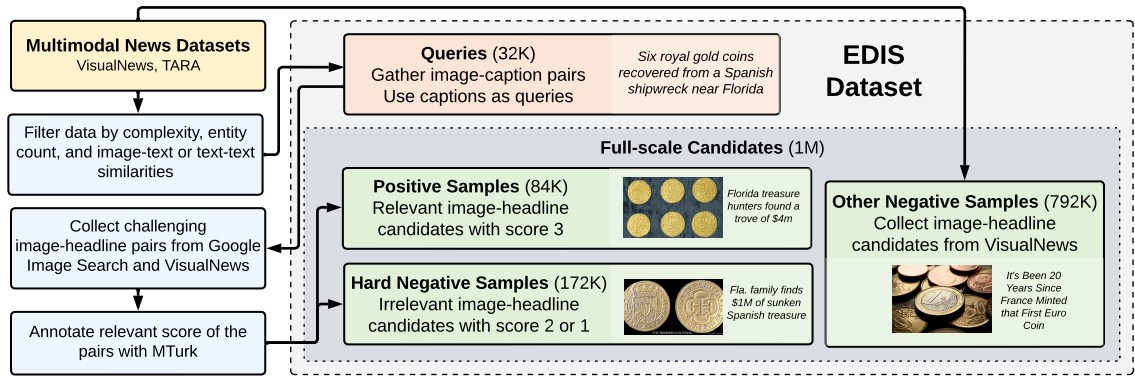

Figure 2: EDIS data collection process. The process consists of query selection, candidate collection and annotation, and hard negative mining.

et al., 2021) have separate encoders for image and text modality and, thus, are much more efficient for retrieval. However, adopting these models for multi-modal candidates leads to architecture modification or suboptimal modality fusion. In contrast, ViSTA (Cheng et al., 2022) can aggregate scene text as an additional input to the candidate encoding branch. In this work, we propose a method named mBLIP to perform feature-level fusion, achieving more effective information fusion for multi-modal candidates.

## 3  Task Formation

EDIS contains a set of text queries $Q = \{q_1, q_2, \ldots\}$, and a set of candidates $c_m$, essentially image-headline pairs $\mathcal{B} = \{c_1 = (i_1, h_1), c_2 = (i_2, h_2), \ldots\}$ where $i_n$ denotes an image and $h_n$ denotes the associated headline. For a query $q_m$, a retrieval model needs to rank top-$k$ most relevant image-headline pairs from $\mathcal{B}$. As is shown in Fig. 1, both images and headlines contain entities that are useful for matching with the query.

We evaluate approaches with both *distractor* and *full* candidate sets. The distractor setup is similar to conventional text-to-image retrieval using MSCOCO (Lin et al., 2014), or Flickr30K (Plummer et al., 2015), where images are retrieved from a limited set $\tilde{\mathcal{B}}$ with ~25K (image, headline) pairs. The full setting requires the model to retrieve images from the entire candidate set $\mathcal{B}$.

## 4  Entity-Driven Image Search (EDIS)

We select queries and candidates from human-written news articles and scraped web pages with different stages of filtering. Then we employ human annotators to label relevance scores. Fig. 2

illustrates the overall dataset collection pipeline.

### 4.1  Query Collection

We extract queries and ground truth images from the VisualNews (Liu et al., 2021), and TARA (Fu et al., 2022) datasets. These datasets contain news articles that have a headline, an image, and an image caption. We adopt captions as text queries and use image-headline pairs as the retrieval candidates.

We design a series of four filters to select high-quality, entity-rich queries. 1) Query complexity: we first evaluate the complexity of queries and remove simple ones with less than ten tokens. 2) Query entity count: we use spaCy to estimate average entity counts in the remaining queries and remove 20% queries with the lowest entity counts. The resulting query set has an average entity count above 4.0. 3) Query-image similarity: to ensure a strong correlation between queries and the corresponding ground truth image, we calculate the similarity score between query-image using CLIP (Radford et al., 2021) and remove 15% samples with the lowest scores. 4) Query-text similarity: we calculate the query-text similarity using Sentence-BERT (Reimers and Gurevych, 2019) and remove the top 10% most similar data to force the retrieval model to rely on visual representations.

To avoid repeating queries, we compare each query to all other queries using BM25 (Robertson et al., 2009). We remove queries with high similarity scores as they potentially describe the same news event and lead to the same retrieved images. As shown in Table 1, we end up with 32,493 queries split into 26K/3.2K/3.2K for train/validation/test.

| | | Retrieval Candidate | | | Text Query | | | |
|---|---|---|---|---|---|---|---|---|
| | # Img | Modality | Source | Label | # Train | # Val | # Test | # Entity |
| Flickr30K (Plummer et al., 2015) | 1K | Image | Flickr | Binary | 145K | 5K | 5K | 0.35 |
| MSCOCO (Lin et al., 2014) | 5K | Image | Flickr | Binary | 566K | 25K | 25K | 0.18 |
| CxC (Parekh et al., 2021) | 5K | Image | Flickr | Continuous | - | 25K | 25K | 0.18 |
| ADE20K (Changpinyo et al., 2021) | 2K | Image | Flickr | Binary | 20K | 2K | - | 0.16 |
| WebQA (Chang et al., 2022) | 390K | Text/Image | Wikimedia | Binary | 34K | 5K | 7.5K | 1.96 |
| EDIS (ours) | **1M** | **Image-Text** | **Open (Google)** | 3-Likert | 26K | 3.2K | 3.2K | **4.03** |

Table 1: Statistics of EDIS and existing image retrieval datasets. EDIS has a larger set of multi-modal candidates, unrestricted image sources, multi-scale annotations, and entity-rich queries compared to previous datasets.

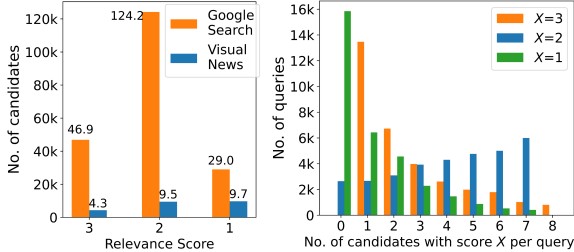

Figure 3: **Left**: Annotated candidates distribution by relevance score. **Right**: Query distribution by the scores of annotated candidates.

## 4.2 Candidate Collection

In web image search experience, multiple relevant images exist for a single query. Therefore, we expand the candidate pool so that each query corresponds to multiple image-headline pairs. Additional candidates are collected from Google Image Search and the rest of the VisualNews dataset. For each query from VisualNews, we select seven image-headline pairs from Google search. For each query from TARA, we select five image-headline pairs from Google search. and two image-headline pairs from VisualNews. Then, we ask annotators to label the relevance score for each candidate on a three-point Likert scale. Score 1 means "not relevant" while 3 means "highly relevant". Formally, denote $\mathbb{E}(\cdot)$ as the entity set and $\mathcal{E}(\cdot)$ as the event of a query $q_m$ or a candidate $c_n = (i_n, h_n)$, we define the ground truth relevance scores as:

$$
rel(m,n) = \begin{cases} 3 & \text{if } \mathbb{E}(q_m) \subseteq \mathbb{E}(c_n) \\ & \text{and } \mathcal{E}(q_m) = \mathcal{E}(c_n), \\ 2 & \text{if } \mathbb{E}(c_n) \cap \mathbb{E}(q_m) \neq \emptyset \\ & \text{and } \mathcal{E}(q_m) = \mathcal{E}(c_n), \\ 1 & \text{if } \mathbb{E}(q_m) \cap \mathbb{E}(i_n) = \emptyset \\ & \text{or } \mathcal{E}(q_m) \neq \mathcal{E}(c_n). \end{cases}
\tag{1}
$$

Each candidate is annotated by at least three workers, and it is selected only when all workers reach a consensus. Controversial candidates that workers cannot agree upon after two rounds of annotations are discarded from the candidate pool. Additionally, one negative candidate is added to each annotation task to verify workers' attention. The final conformity rate among all annotations is over 91.5%.

**Hard Negatives Mining** We discover that EDIS queries can be challenging to Google Image Search in some cases. Among the 200K images from Google Search, 29K ($\sim$15%) are annotated with a score of 1, and 124K ($\sim$62%) are annotated with a score of 2. These candidates are hard negatives that require retrieval models to understand and ground visual entities in the images. As for candidates from VisualNews, there are 9.7K ($\sim$41%) with a score of 1 and 9.5K ($\sim$40%) candidates with a score of 2. We refer to these samples as in-domain hard negatives as their headlines share some entities with the query but refer to different events with discrepant visual representations.

**Soft Negative Mining** Lastly, we utilize the rest of the image-headline pairs from VisualNews and TARA to augment the candidate pool. These candidates are naturally negative candidates with a relevance score of 1 because of the unique article contents and extensive diversity in topics. Therefore, our dataset consists of 1,040,919 image-headline candidates in total.

## 4.3 Dataset Statistics

We demonstrate the major advantage of EDIS over existing datasets in Table 1. EDIS has the largest candidate set with a consistent candidate modality. Our images are not restricted to a specific source as a result of collecting images from a real search engine. Queries from EDIS are entity-rich compared to datasets with general objects (e.g. MSCOCO).

In Fig. 3 (left), we show the score distribution of

human annotations. Candidates mined from Visual News are mostly in-domain hard negatives, while the images represent missing entities or different events. These candidates are mostly annotated with scores of 1 or 2. As for Google search candidates, many images depict the same event but with missing entities. Therefore, the annotations concentrate on score 2. In Fig. 3 (right), we show that most of the queries have at least one hard negative, usually more score 2 negatives than score 1 negatives. About half of the queries have more than one positive candidate (score 3). We show more examples of EDIS candidates in Fig. 8-11.

## 5 Multi-Modal Retrieval Method

Given a text query $q$, a model should be able to encode both images $i_n$ and headlines $h_n$ to match the query encoding. Therefore, the model should include a multi-modal candidate encoder $f_C$ and a query encoder $f_Q$. Within $f_C$, there is a branch for image input $f_I$ and a branch for headline $f_H$. We formalize the matching process between a query $q_m$ and a candidate $c_n = (i_n, h_n)$ as:

$$s_{m,n} = f_Q(q_m)^T f_C(i_n, h_n) \quad (2)$$

where $s_{m,n}$ is the similarity score between $q_m$ and $c_n$. Based on the design of $f_C$, we categorize methods into *score-level fusion* and *feature-level fusion*.

**Score-level Fusion** These methods encode image and headline independently and compute a weighted sum of the features, i.e., $f_C(i_n, h_n) = w_1 f_I(i_n) + w_2 f_H(h_n)$. Therefore, $s_{m,n}$ is equivalent to a weighted sum of query-image similarity $s_{k,k}^i$ and query-headline similarity $s_{k,k}^h$:

$$s_{m,n} = f_Q(q_m)^T (w_1 f_I(i_n) + w_2 f_H(h_n)) \quad (3)$$
$$= w_1 s_{m,n}^I + w_2 s_{m,n}^H \quad (4)$$

Specifically, CLIP (Radford et al., 2021), BLIP (Li et al., 2022), and a combination of models like CLIP and BM25 (Robertson et al., 2009) belong to this category.

**Feature-level Fusion** In Sec. 6, we show that score-level fusion is a compromised choice for encoding multi-modal candidates. Therefore, we propose a modified version of BLIP (mBLIP) to fuse features throughout the encoding process. The overall fusion process can be abstracted as follows:

$$f_C(i_n, h_n) = f_H(h_n, f_I(i_n)) \quad (5)$$
$$s_{m,n} = f_Q(q_m)^T f_H(h_n, f_I(i_n)). \quad (6)$$

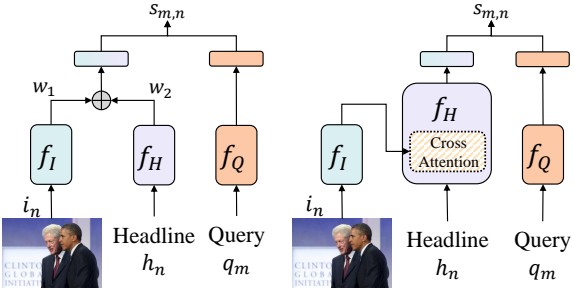

Figure 4: Score-level fusion encodes each modality into a single feature vector, while feature-level fusion outputs a single feature vector for multi-modal candidates by adopting cross-attention layers.

As is shown in Fig. 4, we first extract image embeddings $f_I(\cdot)$ using the image encoder and then feed $f_I(\cdot)$ into the cross-attention layers of $f_H$. The output from $f_H$ is a feature vector $v_{i,h}$ that fuses the information from both image and text modalities. We separately obtain the query feature $v_q = f_Q(q_m)$ where $f_Q$ shares the same architecture and weights with $f_H$, except that the cross-attention layers are not utilized. We adopt the Image-Text Contrastive (ITC) loss (Li et al., 2021) between $v_{i,h}$ and $v_q$ to align the fused features with query features.

## 6 Experiment Setup

### 6.1 Baselines

For score-level fusion mentioned in Sec. 5, we consider CLIP, BLIP, fine-tuned BLIP, and BM25+CLIP reranking to utilize both modalities of the candidates. In addition, we benchmark existing text-to-image retrieval methods, and text document retrieval methods, including VinVL (Zhang et al., 2021), ALBEF (Li et al., 2021), and BM25 (Robertson et al., 2009). Although they are not designed for multi-modal candidates, benchmarking these methods facilitates our understanding of the importance of single modality in the retrieval process. We do not consider single-stream approaches like UNITER (Chen et al., 2020) as they are not efficient for large-scale retrieval and result in extremely long execution time (see Appendix A).

### 6.2 Evaluation Metrics

We evaluate retrieval models with the standard metric **Recall@$k$** (R@$k$) that computes the recall rate of the top-$k$ retrieved candidates. $k$ is set to $1, 5, 10$. We report mean Average Precision (**mAP**) to reflect the retrieval precision considering the ranking

position of all relevant documents. Formally,

$$\text{Recall@}k = \frac{1}{|Q|} \sum_{m=1}^{|Q|} \frac{\sum_{n=1}^{k} \overline{rel}(m,n)}{\sum_n \overline{rel}(m,n)} \quad (7)$$

$$\text{mAP} = \frac{1}{|Q|} \sum_{m=1}^{|Q|} \frac{\sum_n P(m,n)\overline{rel}(m,n)}{\sum_n \overline{rel}(m,n)} \quad (8)$$

$$\overline{rel}(m,n) = \begin{cases} 1 \text{ if } rel(m,n) = 3 \\ 0 \text{ otherwise} \end{cases} \quad (9)$$

where $P(m,n)$ is the Precision@$n$ of a query $q_m$. For R@$k$ and mAP, candidates with relevant score 3 are positive candidates, while candidates with scores 2 or 1 are (hard) negative samples. These two metrics reflect the model's ability to retrieve the most relevant candidates, which aligns with the definition in Fig. 2.

To give merits to candidates with a score of 2, we also report Normalized Discounted Cumulative Gain (**NDCG**). NDCG assigns importance weights proportional to the relevance score so that ranking score 2 candidates before score 1 candidate will lead to a higher metric value.

$$DCG(m) = \sum_n \frac{rel(m,n) - 1}{log_2(1 + n)} \quad (10)$$

$$NDCG = \frac{1}{|Q|} \sum_{m=1}^{|Q|} \frac{DCG(m)}{IDCG(m)}, \quad (11)$$

where $IDCG(m)$ is the $DCG$ value of $q_m$ with the ideal candidate ranking.

### 6.3 Implementation Details

For BLIP fine-tuning, we adopt the same loss and hyperparameters as reported in the original implementation. We increase the learning rate to 1e-5 for optimal validation results. We directly rank the candidates by computing the cosine similarity of query features and candidate features and do not use any linear regression heads for reranking. Therefore, we abandon the image-text matching (ITM) loss in mBLIP fine-tuning and increase the learning rate to 5e-5 for optimal performance. More details can be found in Appendix A.

## 7 Experimental Results

### 7.1 BLIP-based fusion methods

We first investigate the performance difference between score-level and feature-level fusion as mentioned in Sec. 5. We implement these two ap-

| | Methods | Fine-tuned | R@1 | R@5 | R@10 | mAP | NDCG |
|---|---|---|---|---|---|---|---|
| Distractor | BLIP | ✗ | 18.4 | 46.6 | 59.0 | 37.8 | 58.1 |
| | BLIP | ✓ | **32.6** | 62.0 | 72.1 | 53.8 | 67.3 |
| | mBLIP | ✓ | 27.8 | **66.0** | **81.4** | **56.2** | **75.3** |
| Full | BLIP | ✗ | 5.8 | 17.3 | 24.0 | 13.4 | 32.4 |
| | BLIP | ✓ | **13.1** | 29.6 | 37.6 | 23.4 | 38.9 |
| | mBLIP | ✓ | 12.3 | **33.3** | **44.1** | **27.4** | **47.9** |

Table 2: Retrieval performance of BLIP and mBLIP under the distractor and full settings. We use grid search on the validation split to find the best score fusion weights (see Eq. 4) for zero-shot and fine-tuned BLIP.

proaches on BLIP (Li et al., 2022). Table 2 compares the result under two different setups where "BLIP" denotes the score-level fusion using the original BLIP architecture, and "mBLIP" denotes our proposed feature-level fusion. For score-level fusion, we obtain the weights from a grid search on the validation set under the distractor setup.

**Distractor Set** Pre-trained BLIP achieves 18.4 R@1 and 46.6 R@5, which means that almost one-third of the queries have a positive candidate in top-1 results, and around half of the positive candidates are retrieved in top-5 results. After fine-tuning, BLIP doubles R@1 to 32.6 and achieves significant gain in other metrics. The improvement shows that entities in EDIS are out-of-domain concepts for zero-shot BLIP, and EDIS training split is useful for models to adapt to the news domain. mBLIP outperforms BLIP in all metrics except R@1. The overall improvement entails that feature-level fusion is superior to score-level fusion by utilizing headlines more effectively. The degradation in R@1 can be attributed to the fact that the image-query alignment is accurate enough for a small number of queries. Therefore, utilizing headlines slightly harms the results as they only provide high-level summarization.

**Full Set** Retrieving from the full candidate set significantly degrades the performance by over 50% in all metrics. Though the distractor setup was widely adopted in previous work, we show that a larger candidate set imposes remarkable challenges to the SOTA models. We can observe similar trends by comparing the three variants of BLIP. mBLIP achieves over 17% relative improvement across all metrics except R@1, even more significant than 4-12% relative improvement under the distractor set. The degradation in R@1 is also much less severe. Therefore, feature-level fusion is a more effective

| | Methods | Fine-tuned | $f_H$ | $f_I$ | EDIS Val | | | | | EDIS Test | | | | |
|---|---|---|---|---|---|---|---|---|---|---|---|---|---|---|
| | | | | | R@1 | R@5 | R@10 | mAP | NDCG | R@1 | R@5 | R@10 | mAP | NDCG |
| | Upper Bound | | | | 60.1 | 97.1 | 100 | 100 | 100 | 61.0 | 97.3 | 100 | 100 | 100 |
| Distractor | BM25 | ✗ | ✓ | ✗ | 6.7 | 22.2 | 29.5 | 20.0 | 54.1 | 6.9 | 21.7 | 28.8 | 19.4 | 52.9 |
| | CLIP | ✗ | ✓ | ✗ | 8.1 | 28.6 | 40.5 | 25.9 | 58.7 | 8.2 | 27.7 | 39.1 | 24.8 | 57.5 |
| | BM25+CLIP | ✗ | ✓ | ✓ | 13.4 | 38.4 | 46.8 | 33.5 | 64.3 | 13.5 | 37.7 | 45.9 | 32.5 | 63.5 |
| | CLIP | ✗ | ✓ | ✓ | 30.7 | **68.0** | 80.5 | **58.2** | 73.8 | 30.1 | **68.1** | 80.2 | **57.2** | 73.3 |
| | BLIP | ✓ | ✓ | ✓ | **32.0** | 62.1 | 72.2 | 53.9 | 67.7 | **32.6** | 62.0 | 72.1 | 53.8 | 67.3 |
| | mBLIP | ✓ | ✓ | ✓ | 27.4 | 65.9 | **81.5** | 57.0 | **75.8** | 27.8 | 66.0 | **81.4** | 56.2 | 75.3 |
| Full | BM25 | ✗ | ✓ | ✗ | 4.7 | 13.3 | 17.3 | 11.9 | 39.4 | 4.7 | 12.9 | 16.6 | 11.4 | 38.0 |
| | CLIP | ✗ | ✓ | ✗ | 4.6 | 13.9 | 18.6 | 12.6 | 38.9 | 5.0 | 13.4 | 18.2 | 12.4 | 28.2 |
| | BM25+CLIP | ✗ | ✓ | ✓ | 6.8 | 18.8 | 23.6 | 16.6 | 45.9 | 6.3 | 18.0 | 22.7 | 15.9 | 45.0 |
| | CLIP | ✗ | ✓ | ✓ | **14.1** | **35.4** | **45.7** | 28.3 | 45.9 | **13.7** | **36.0** | **47.2** | **28.0** | 46.0 |
| | BLIP | ✓ | ✓ | ✓ | 12.3 | 29.0 | 37.8 | 22.9 | 38.8 | 13.1 | 29.6 | 37.6 | 23.4 | 38.9 |
| | mBLIP | ✓ | ✓ | ✓ | 13.3 | 34.1 | 44.3 | **28.9** | **49.0** | 12.3 | 33.3 | 44.1 | 27.4 | **47.9** |

Table 3: Evaluation results on additional baselines.

way to encode multi-modal candidates, considering that users usually receive more than one searched image in reality.

## 7.2 Additional Baselines

In Table 3, the defective recall rates of BM25 and CLIP text encoder imply that headlines solely are insufficient for accurate retrieval. However, text-based retrieval achieves promising NDCG values, indicating that headlines are useful for ranking score 2 candidates to higher positions.

**Score-level Fusion** "BM25+CLIP" first ranks candidates using BM25 and then reranks the top 50 or 200 candidates with CLIP to utilize the images. Despite the improvement compared to text-based methods, it underperforms zero-shot CLIP or BLIP. This implies that ranking with query-headline similarity imposes a bottleneck on the reranking process. CLIP achieves the best performance in terms of R@1/5/10 and mAP compared to other methods. We hypothesize that the "CLIP filtering" step in Sec. 4.1 eliminates hard negative query-image pairs for CLIP and thus introduces performance bias towards CLIP. Fine-tuned CLIP does not show apparent improvement and thus is not shown in Table 3. Therefore, EDIS is still challenging for SOTA retrieval models.

**Feature-level Fusion** mBLIP consistently outperforms other approaches in NDCG regardless of the candidate set scale, achieving 75.3/47.9 NDCG under distractor/full set. mBLIP fuses headline features with visual features more effectively and thus proves that headlines are critical for ranking

| | Methods | $f_H$ | $f_I$ | EDIS Test | | | | |
|---|---|---|---|---|---|---|---|---|
| | | | | R@1 | R@5 | R@10 | mAP | NDCG |
| Distractor | BLIP | ✓ | ✗ | 6.6 | 21.6 | 30.5 | 19.7 | 51.9 |
| | BLIP | ✗ | ✓ | 33.9 | 61.2 | 71.3 | 54.0 | 66.5 |
| | BLIP | ✓ | ✓ | 32.6 | 62.0 | 72.1 | 53.8 | 67.3 |
| | mBLIP | ✓ | ✗ | 8.8 | 29.1 | 40.1 | 26.0 | 57.9 |
| | mBLIP | ✗ | ✓ | 22.2 | 49.9 | 60.2 | 40.9 | 58.2 |
| | mBLIP | ✓ | ✓ | 27.8 | 66.0 | 81.4 | 56.2 | 75.3 |

Table 4: Ablation study on the effectiveness of feature fusion in BLIP and mBLIP.

score 2 candidates higher. We conjecture that many score 2 images have insufficient entities, resulting in lower query-image similarity scores. Hence, models must rely on headlines to simultaneously recognize entities from multiple modalities.

## 7.3 Ablation Study

**Component Analysis** Table 4 shows the performance of two fusion approaches without either image or headline branch. BLIP achieves much lower performance when relying solely on query-headline alignment (6.6 R@1, 29.7 mAP) compared to utilizing images only (33.9 R@1, 54.0 mAP). BLIP only achieves comparable and slightly degraded performance when using images and headlines for score fusion. Therefore, score-level fusion cannot easily tackle multi-modal candidates in EDIS.

In contrast, mBLIP shows improved performance with the headline encoder while decreased performance with the image encoder only. This is intuitive as the BLIP fine-tuning process only utilizes images without headlines, yet mBLIP utilizes both images and headlines. More interestingly,

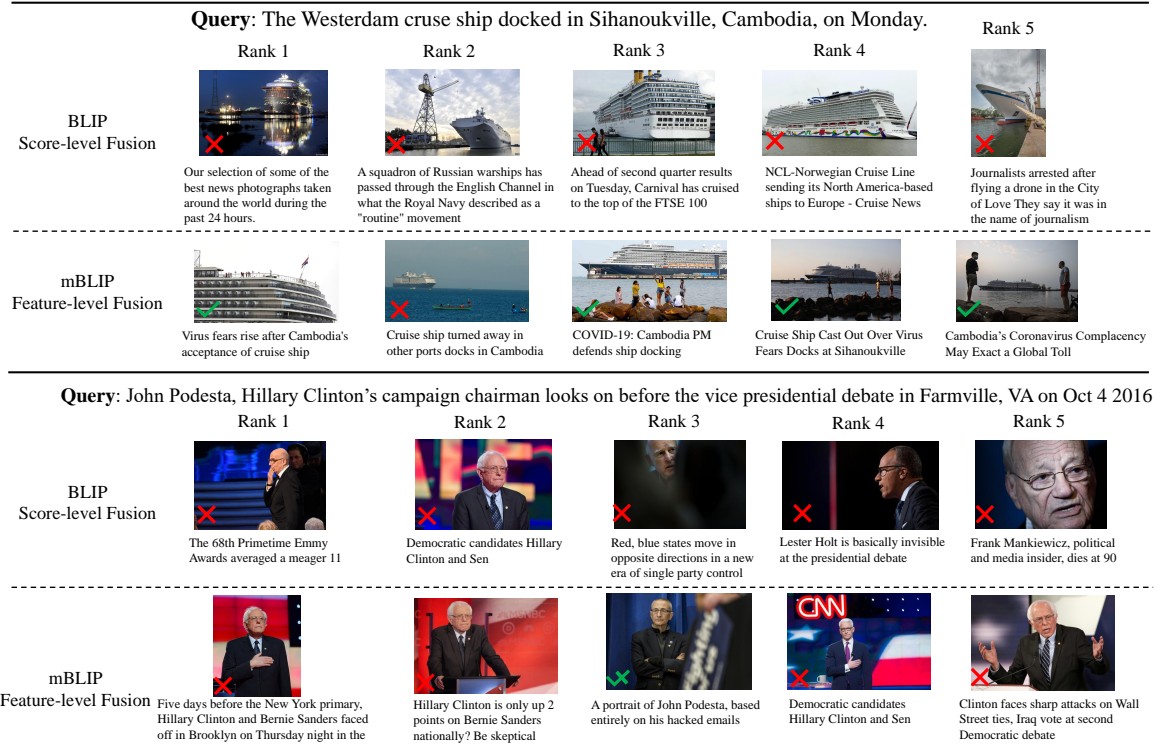

Figure 5: A success case (top) and a failure case (bottom) of mBLIP compared to BLIP.

when using both image and headline encoders, mBLIP demonstrates over 20% relative increase in all metrics. The results imply that feature-level fusion is a more effective method to combine candidate features from multiple modalities.

## 7.4 Case Study

**Success Case** We show one success case and one failure case of mBLIP in Fig. 5. In the success case (top), mBLIP manages to retrieve all four relevant images while BLIP retrieves five false positives. Since all ten images contain a "cruise", we conjecture that entities in headlines (e.g., "Cambodia", "Sihanoukville") play a critical role for mBLIP to outperform BLIP in this case. The case shows feature-level fusion is much more effective in utilizing headline features than score-level fusion.

**Failure Case** As for the failure case in Fig. 5 (bottom), BLIP and mBLIP fail to retrieve the positive candidates in the top-5 results. Both methods fail to recognize "John Podesta" and align the text with the visual representation. For example, the top-2 candidates retrieved by mBLIP depict a different person from a different event. "Hillary Clinton" becomes a distracting entity in the query, and the model must understand the event instead of just matching entities to achieve accurate retrieval

results. The third candidate of mBLIP shows the image with the correct person but from a different event. It further proves that the EDIS is a challenging dataset that requires specific knowledge of entities, cross-modal entity matching, and event understanding.

## 8 Conclusion

Training and evaluating large-scale image retrieval datasets is an inevitable step toward real image search applications. To mitigate the gap between existing datasets and real-world image search challenges, we propose a large-scale dataset EDIS with a novel retrieval setting and one million candidates. EDIS queries and candidates are collected from the news domain describing abundant entities and events. EDIS candidates are image-headline pairs since realistic image search utilizes the surrounding text of an image to facilitate accurate searching results. As a primary step towards handling multimodal candidates in EDIS, we review two primary fusion approaches and propose a feature-level fusion method to utilize the information from both images and headlines effectively. Our experimental results show ample space for improvement on EDIS. Future work should consider more principled solutions involving knowledge graphs, entity

linking, and training algorithm design.

## 9  Acknowledgement

The work was funded by the National Science Foundation award #2048122. The writers' opinions and conclusions in this publication are their own and should not be construed as representing the sponsors' official policy, expressed or inferred.

## 10  Limitations

In this study, we only cover image retrieval datasets with English instructions. Queries and headlines in other languages may characterize different types of ambiguity or underspecification. Thus, expanding the datasets to multi-lingual image retrieval based on our dataset is important. Secondly, we only consider the news domain to collect entity-rich queries and images. We plan to expand our dataset to open-domain where other entities like iconic spots will be included. In addition, we only consider the headlines as the text information to utilize in the retrieval process. However, in real image search scenarios, search engines usually utilize multiple paragraphs of the surrounding text to determine the relevance of the image. In the future, we will expand the text of the multimodal candidates with news articles or segments of the articles. Our dataset and models trained on it could be biased if the model is not accurate enough. The model may return completely incorrect candidates and cause users to confuse persons or objects with incorrect identities. We will provide all ground truth annotations with visualization code to help users learn about the ground truth candidates. Last but not least, we do not consider the phenomenon of underspecification in the image search experience. Users search with phrases or incomplete sentences to save typing efforts. Therefore, more realistic queries can be underspecified and grammatically incorrect. However, this is a problem universal to all existing image retrieval datasets, as collecting real human search results could be challenging. We plan to make our dataset more realistic in the future by utilizing powerful tools such as large language models to generate underspecified, near-realistic queries.

## 11  Ethics Consideration

We will release our dataset EDIS for academic purposes only and should not be used outside of research. We strictly follow any licenses stated in the datasets that we have newly annotated. As introduced in Sec. 4.2, we annotated the data with crowd-workers through Amazon Mechanical Turk. The data annotation part of the project is classified as exempt by our Human Subject Committee via IRB protocols. We required the workers to be in English-speaking regions (Australia, Canada, New Zealand, the United Kingdom, and the United States). We keep the identity of workers anonymized throughout the collection and post-processing stages. We also require the workers to have a HIT approval rating of $\geq 96\%$ or higher. We pay each completed HIT \$0.2, and each HIT takes around 30-40 seconds to complete on average. Therefore, this resulted in an hourly wage of \$18-\$24, as determined by the estimation of completing time for each annotation task. Example screenshots of our annotation interface can be found in Fig. 6-7 under Appendix A.

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

# A Appendix

## A.1 Additional Implementation Details

**Hyperparameters** We fine-tuned BLIP and mBLIP on 4 40GB NVIDIA A100. It takes 5 hours for BLIP fine-tuning and 3 hours for mBLIP. For both BLIP and mBLIP, we train the model for 6 epochs with batch size 16 per GPU. The model checkpoint with the best recall rate over the validation set is selected for final evaluation. We apply grid search for score-level fusion using BLIP or CLIP to find the optimal $w_1$. We first search over ten intermediate numbers between 0 and 1 and then narrow the range to search for 100 intermediate numbers. Finally, we found training and validation results stable without much randomness for all implemented methods. Therefore, we evaluate every model once and report the metric values of one-time evaluation.

**BLIP Training** For BLIP fine-tuning, we follow the original implementation and adopt the original image-text contrastive (ITC) loss and image-text matching (ITM) loss. We only utilize images with scores of 3 and text queries for training. As for mBLIP, the headline encoder and the query encoder share the same weight. We utilize images with scores of 3, associated headlines, and text queries for training. The output from the image encoder is fed into the transformer layers of the headline encoder through cross-attention layers. Then the output of the headline encoders can be treated as the fused feature of image-headline pairs. We compute ITC loss based on the headline encoder outputs and the query encoder outputs.

**Single Stream Models** We do not evaluate any single stream models or modules due to time complexity. Consider $m$ queries and $n$ candidates. The complexity for a dual encoder model to obtain all features is $O(m+n)$. The computation cost of computing cosine similarity is trivial compared to the forward process of a model and can be neglected. However, for a single stream model, it takes $O(mn)$

to obtain similarity scores for all query-candidate pairs. Since it takes around 3.5 minutes for BLIP to evaluate over 3.2k queries with 25K candidates, it is taking more than 5 days for a single encoder model to complete retrieval under the distractor setting. It takes more than a year to complete retrieval under the full setting.

You are given a Query and eight image-text samples. Your task is to determine if each image-text matches the Query.

- Please read the Query carefully.
- For each image-text sample, give a score on how much it matches the **Query**.
- Focus on the images and use text to understand the background information.

**How to rate a score:**
- **3: EVERYTHING about the image is CORRECT, such as subject, event, location, and action. Be strict.**
- **2: Image is relevant to the query, but some entities are missing or incorrect.**
- **1: Image is incorrect or missing the semantic meaning of the query.**

Notice: There may be additional false samples with an expected score of 1 inserted to verify your attention.

Take a look at the two examples below.

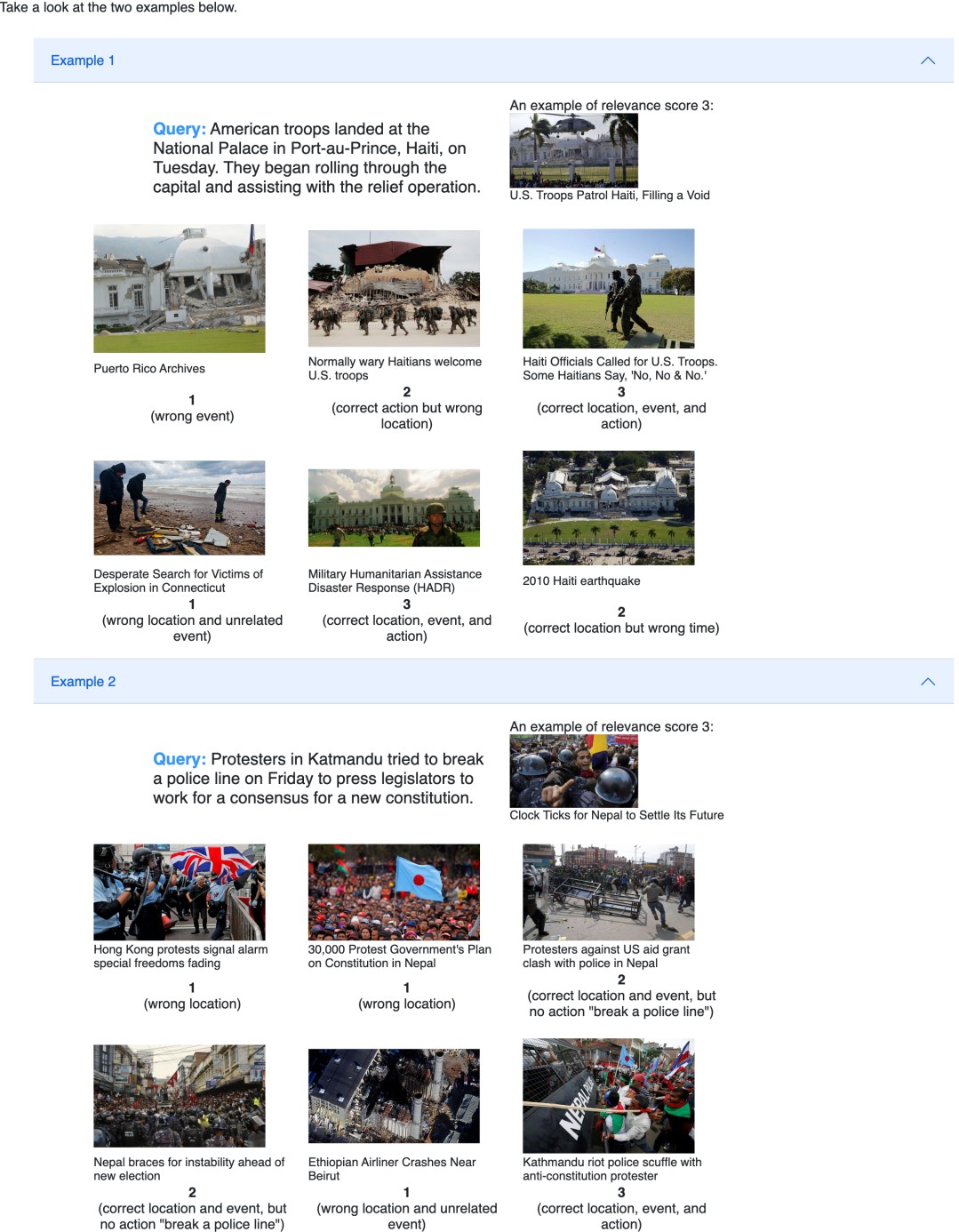

Figure 6: Amazon Mechanical Turk interface with annotation examples for crowd-workers to read

You are given a Query and eight image-text samples. Your task is to determine if each image-text matches the Query.

- Please read the Query carefully.
- For each image-text sample, give a score on how much it matches the **Query**.
- Focus on the images and use text to understand the background information.

**How to rate a score:**

- **3: EVERYTHING about the image is CORRECT, such as subject, event, location, and action. Be strict.**
- **2: Image is relevant to the query, but some entities are missing or incorrect.**
- **1: Image is incorrect or missing the semantic meaning of the query.**

Notice: There may be additional false samples with an expected score of 1 inserted to verify your attention.

Take a look at the two examples below.

| Example 1 | ˅ |
|---|---|
| Example 2 | ˅ |

Complete the following:

**Query:** Audiences react to standup comedians performing an act called AisiTaisi Democracy at the Canvas Laugh Club at The Palladium Mall in Mumbai

An example of relevance score 3:

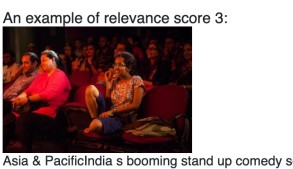

Asia & PacificIndia s booming stand up comedy scene tests boundaries with cutting edge jokes

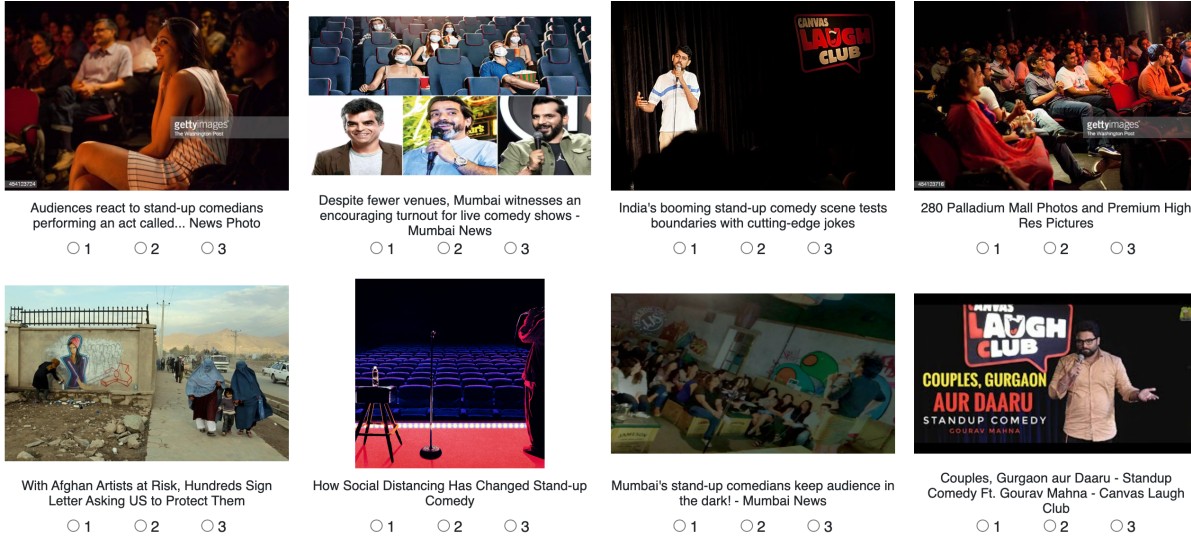

Audiences react to stand-up comedians performing an act called... News Photo
○ 1    ○ 2    ○ 3

Despite fewer venues, Mumbai witnesses an encouraging turnout for live comedy shows - Mumbai News
○ 1    ○ 2    ○ 3

India's booming stand-up comedy scene tests boundaries with cutting-edge jokes
○ 1    ○ 2    ○ 3

280 Palladium Mall Photos and Premium High Res Pictures
○ 1    ○ 2    ○ 3

With Afghan Artists at Risk, Hundreds Sign Letter Asking US to Protect Them
○ 1    ○ 2    ○ 3

How Social Distancing Has Changed Stand-up Comedy
○ 1    ○ 2    ○ 3

Mumbai's stand-up comedians keep audience in the dark! - Mumbai News
○ 1    ○ 2    ○ 3

Couples, Gurgaon aur Daaru - Standup Comedy Ft. Gourav Mahna - Canvas Laugh Club
○ 1    ○ 2    ○ 3

Figure 7: Amazon Mechanical Turk interface and annotation task

**Query:** Six royal gold coins recovered from a Spanish shipwreck near Florida

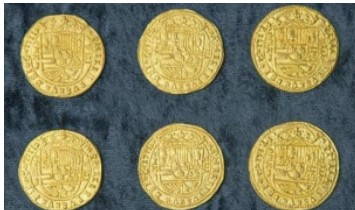

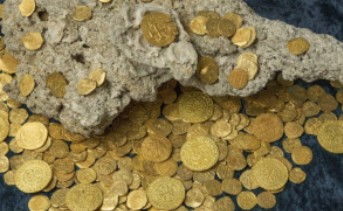

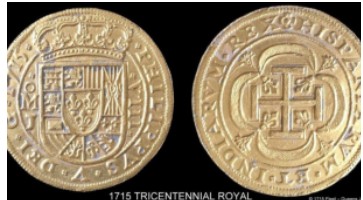

**Text:** Florida treasure hunters found a trove of $4m
**Score:** 3

**Text:** Florida treasure hunters find $4.5m in rare Spanish coins - Florida
**Score:** 2

**Text:** Fla. family finds $1M of sunken Spanish treasure
**Score:** 2

---

**Query:** Police vehicles in front of Deutsche Bank headquarters in Frankfurt on Thursday. Prosecutors raided the companys office in a case related to the Panama Papers

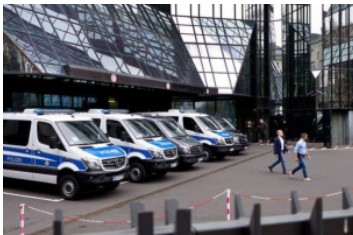

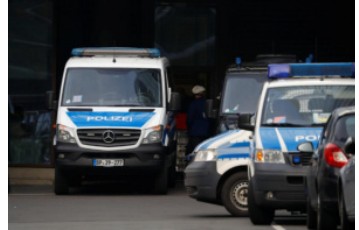

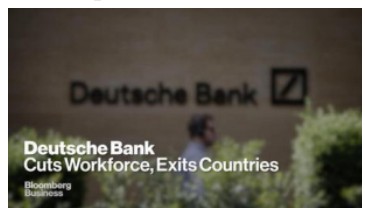

**Text:** Deutsche Bank Offices Are Searched in Money Laundering Investigation
**Score:** 3

**Text:** Police raid Deutsche Bank headquarters as part of Panama Papers money laundering investigation
**Score:** 2

**Text:** Deutsche Bank said Thursday it would shed 35,000 jobs and close operations in 10 countries
**Score:** 1

---

**Query:** Barack Obama accompanied by first lady Michelle was making his first visit to Fort Bragg

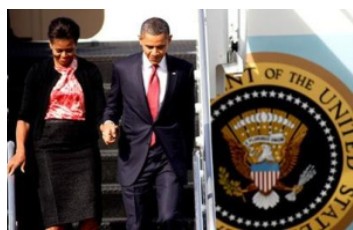

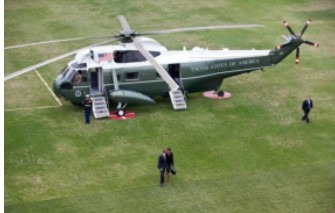

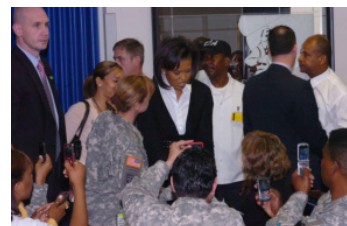

**Text:** President Barack Obama marked the end of the Iraq war with a tribute to the troops who fought and died in a conflict he opposed from the start
**Score:** 3

**Text:** President Obama and First Lady Michelle Obama Return from Ft. Bragg
**Score:** 2

**Text:** First Lady visits Fort Bragg, vows support for military families
**Score:** 2

Figure 8: Additional EDIS dataset examples set 1

**Query:** A rally in front of the New Jersey State House on Thursday, when the authorization for the state's transportation trust fund expired

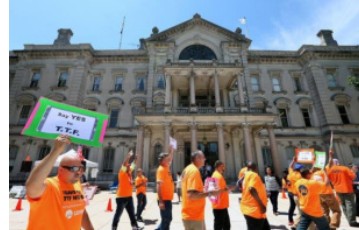 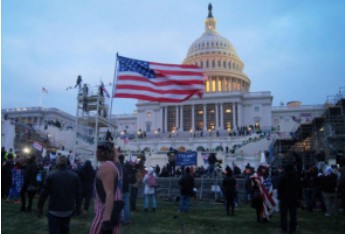 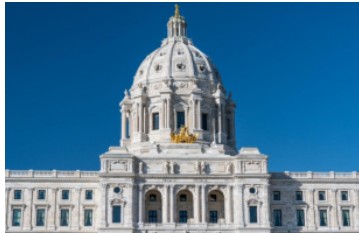

**Text:** With No Deal on New Jersey Gas Tax, Christie Orders Shutdown of Road Projects
**Score:** 3

**Text:** Aftermath of the 2021 United States Capitol attack
**Score:** 1

**Text:** TEAMSTERS 320
**Score:** 1

---

**Query:** Unloading goods in Lianyungang, China. The United States trade deficit with China soared in the first three months of this year

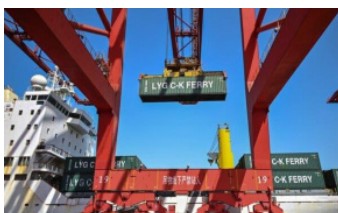 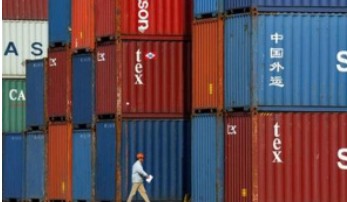 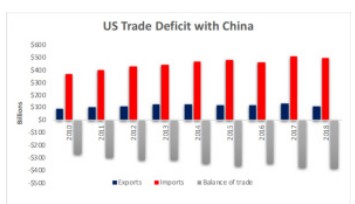

**Text:** A Temporary U.S.-China Trade Truce Starts to Look Durable
**Score:** 3

**Text:** The US trade deficit with China has hit a record high, fuelling tensions between the countries over currency imbalances
**Score:** 2

**Text:** Will China Reverse Its Trade Surplus with the United States
**Score:** 1

---

**Query:** Femke Van Den Driessche during the world cyclocross championships in HeusdenZolder on Saturday A concealed motor was later found on her bicycle

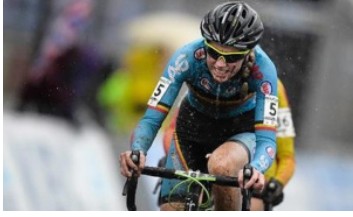

**Text:** Chris Froome has said he raised his concerns over motorised doping to the authorities a year ago
**Score:** 3

**Text:** UCI confirms motorised bike at Cyclo-cross World Championships
**Score:** 2

**Text:** Controversy as motor found inside bike at the world cyclo-cross championships
**Score:** 2

**Query:** Diane Keaton wins her Oscar for 1977 s Annie Hall

Figure 9: Additional EDIS dataset examples set 2

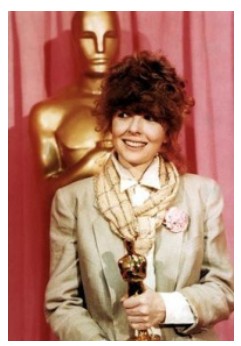 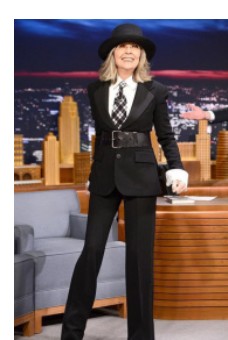 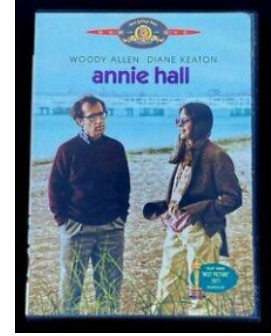

**Text:** The actress Greta Gerwig surely deserves to receive the keys to New York in recognition of her on screen celebrations of that city
**Score:** 3

**Text:** Diane Keaton Says It's All Because of 'Annie Hall'
**Score:** 2

**Text:** Annie Hall DVD 1977 Woody Allen Diane Keaton Oscar Winner Best Picture LIKE NEW
**Score:** 2

---

**Query:** A dog was rescued last month from a farm in Wonju, South Korea. Humane Society International offers to pay farmers to release dogs so they can be sent abroad to be adopted

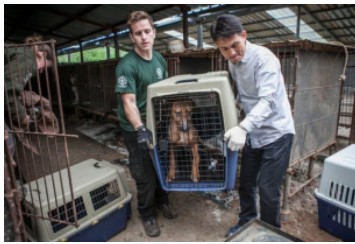 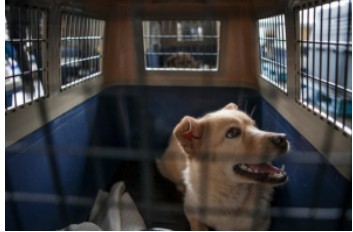 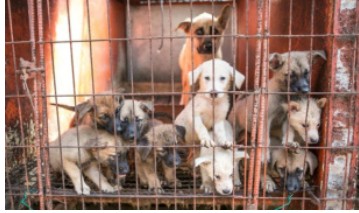

**Text:** From Dog Farms in South Korea to New Lives as Pets Abroad
**Score:** 3

**Text:** From Dog Farms in South Korea to New Lives as Pets Abroad
**Score:** 3

**Text:** Flagler Humane Society relates story of South Korean dogs arriving in US
**Score:** 2

---

**Query:** A Hubble Telescope view reveals Uranus surrounded by its four major rings and 10 of its 17 known satellites

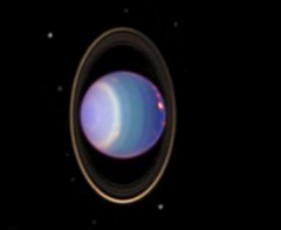 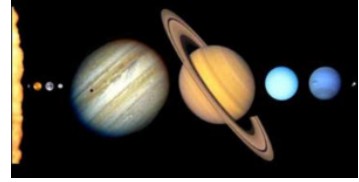 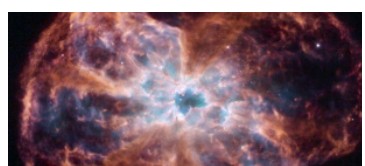

**Text:** Speaking of ScienceHubble captures a shimmering butterfly in space
**Score:** 3

**Text:** Hubble Finds Rings In Uranus Orbit
**Score:** 2

**Text:** Scientists Revisit Old Data, Discover New Moons Around Uranus
**Score:** 1

Figure 10: Additional EDIS dataset examples set 3

**Query:** Matthew Centrowitz won the first US gold in men s 1500 since 1908

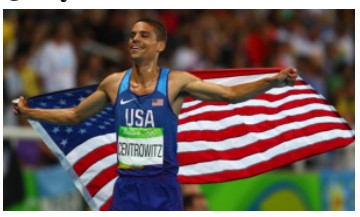

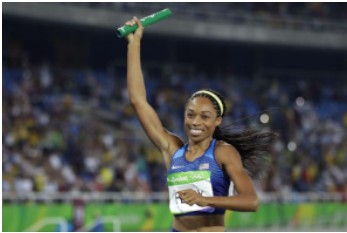

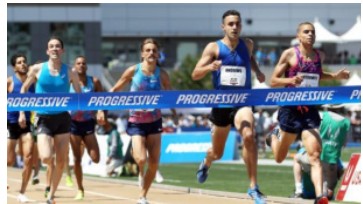

**Text:** Matt Centrowitz wins gold in men's 1,500; first for USA in 108 years
**Score:** 3

**Text:** Matt Centrowitz Takes 1,500; First U.S. Winner Since 1908 : The Torch : NPR
**Score:** 2

**Text:** Robby Andrews upsets Olympic gold medalist Matt Centrowitz in 1,500 at U.S. championships
**Score:** 2

---

**Query:** Federal Reserve Chair Janet L Yellen testifies Wednesday before the House Financial Services Committee in Washington

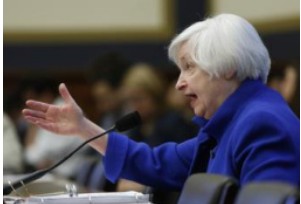

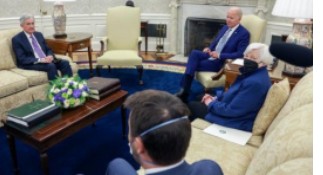

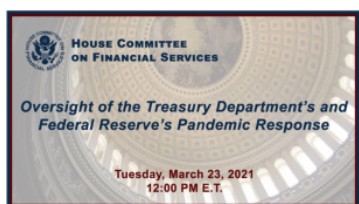

**Text:** WonkblogLawmakers grill Fed chief in testy hearing on Capitol Hill
**Score:** 3

**Text:** Andrew Ross Sorkin: Janet Yellen, Fed 'got it wrong' on inflation, were 'a little political' in their response
**Score:** 2

**Text:** Virtual Hearing - Oversight of the Treasury Department's and Federal Reserve's Pandemic Response
**Score:** 1

---

**Query:** Ben Fogle holds an Olympic torch at the Eden Project near Bodelva Cornwall

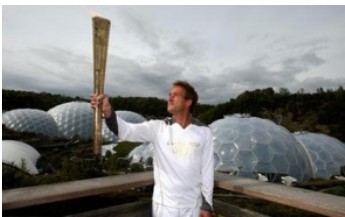

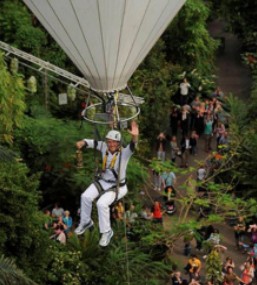

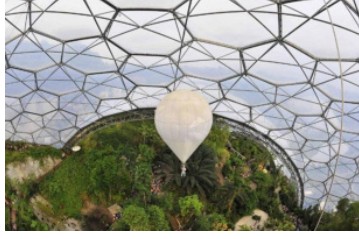

**Text:** It was meant to be a chance for ordinary people across Britain to inspire others with the Olympic spirit
**Score:** 3

**Text:** TV adventurer Ben Fogle set to swim the Atlantic
**Score:** 2

**Text:** Olympic flame goes out in relay
**Score:** 1

Figure 11: Additional EDIS dataset examples set 4