# OpenReview forum: "EDIS: Entity-Driven Image Search over Multimodal Web Content"
_EMNLP/2023/Conference — EMNLP 2023 Main_

### Official Review · Reviewer_QYbK · 2023-08-06

**Soundness:** 4

**Excitement:**

3: Ambivalent: It has merits (e.g., it reports state-of-the-art results, the idea is nice), but there are key weaknesses (e.g., it describes incremental work), and it can significantly benefit from another round of revision. However, I won't object to accepting it if my co-reviewers champion it.

**Paper Topic And Main Contributions:**

This paper proposes a new retrieval task from textual query to the pair consisting of an image and corresponding headline. To implement this, the authors modified the architecture of BLIP to allow the fusion of image and headline. Furthermore, they propose a large-scale dataset to evaluate the task.

**Reasons To Accept:**

- This paper proposed a new task to enhance the entity recognition
- A new dataset is collected which can facilitate future research and evaluation
- Experiments are conducted on the dataset to show the

**Reasons To Reject:**

This paper claims three expectations in the abstract. However, the task formulation and the collected dataset cannot evaluate these aspects properly. For example, how to evaluate the understanding of the entity name? Is it grounded in an image or simply in the headline? In addition, the news corpus is usually of high quality in terms of both format and expression. However, the real application includes much more noisy web data.

**Reproducibility:**

5: Could easily reproduce the results.

**Reviewer Confidence:**

3: Pretty sure, but there's a chance I missed something. Although I have a good feel for this area in general, I did not carefully check the paper's details, e.g., the math, experimental design, or novelty.

---

> ### Author Rebuttal · Authors · 2023-08-28
>
> Thank you for your valuable feedback and comments.
>
> > How to evaluate the understanding of the entity names?
>
> Please note that understanding the entity names in the query requires both grounding to the images, matching entities and understanding events in the headlines. For example, in the Fig. 1 Rank 2 image, EDIS requires the model to ground Rick Santorum to the visual figures and understand that Rick Santorum is missing from the image. In the Rank 4 headline, EDIS requires the retrieval model to understand that “the New Hampshire Republican debate” in the query is different from the “Final GOP Debate on CNN” in the headline.
>
> For now, the understanding of entity names is implicitly reflected in the reported Recall, mAP, and NDCG metrics. A higher metric score usually represents a better understanding of the named entities and grounding abilities. A possible direct evaluation would be masking out each of the named entities in the text query, and computing the change in metrics scores to see the influence of each entity on the retrieval results.
>
>
> > News corpus is of high quality while real applications includes noisy web data.
>
> While the real web environments could consist of noisy web data, we are not targeting at the problem of multi-modal retrieval under noisy data or how to learn from noisy data to perform accurate retrieval. Our work mainly addresses the challenges in understanding named entities and events and grounding them to multi-modal candidates in a large-scale. The introduced high-level headlines are challenging for existing models to effectively fuse the information with the image modality. Thank you for your suggestions and we will consider more noisy web data as a future work.

---

### Official Review · Reviewer_q3bm · 2023-08-06

**Typos Grammar Style And Presentation Improvements:** Line 405
**Soundness:** 4

**Excitement:**

4: Strong: This paper deepens the understanding of some phenomenon or lowers the barriers to an existing research direction.

**Missing References:**

I didn't identify any missing references.

**Paper Topic And Main Contributions:**

In this paper, the authors propose a dataset called EDIS with 1 million images for cross-modal image search in the news domain.
EDIS dataset includes images and text with entities and events. Moreover, the authors propose a feature-level fusion method for multi-modal inputs. Existing approaches are evaluated on the new dataset.


**Reasons To Accept:**

The paper is well written and it contributes a useful artifact, i.e., the EDIS dataset.


**Reasons To Reject:**

I don't see any reason to reject this paper.

**Reproducibility:**

4: Could mostly reproduce the results, but there may be some variation because of sample variance or minor variations in their interpretation of the protocol or method.

**Reviewer Confidence:**

3: Pretty sure, but there's a chance I missed something. Although I have a good feel for this area in general, I did not carefully check the paper's details, e.g., the math, experimental design, or novelty.

---

> ### Author Rebuttal · Authors · 2023-08-28
>
> Thank you for your valuable comments and acknowledgment of our work. Please let us know if you have any questions or concerns. We are always happy to address them in the discussions.

---

### Official Review · Reviewer_eaas · 2023-08-11

**Typos Grammar Style And Presentation Improvements:** N/A
**Soundness:** 3

**Excitement:**

3: Ambivalent: It has merits (e.g., it reports state-of-the-art results, the idea is nice), but there are key weaknesses (e.g., it describes incremental work), and it can significantly benefit from another round of revision. However, I won't object to accepting it if my co-reviewers champion it.

**Missing References:**

N/A

**Paper Topic And Main Contributions:**

This paper is about Entity-driven image search topics, and the main contributions of this paper are:

1. They collect and annotate EDIS for large-scale image search, which characterizes single-modality queries and multi-modal candidates.
2. They propose a feature-level fusion method for multi-modal inputs before measuring alignment with query features.
3. They evaluate several existing methods on the new EDIS dataset.

**Questions For The Authors:**

- will you release the new collected EDIS dataset?
- Please refer to the previous section.

**Reasons To Accept:**

The paper collected a new large-scale dataset EDIS, and evaluated several existing methods.

**Reasons To Reject:**

- The author only proposed a modified BLIP method which is a minor modification. Based on the results of Table 3, we found that the best results usually come from CLIP and BLIP, not their proposed new method. It looks like their proposed method does not always work.
- on Table 4, we can find that BLIP perform better than mBLIP in two situations. It also demonstrates my previous conclusion.
- I think the authors can also evaluate their proposed method on MS-COCO and Flikr30K with other SOTAs.

**Reproducibility:**

3: Could reproduce the results with some difficulty. The settings of parameters are underspecified or subjectively determined; the training/evaluation data are not widely available.

**Reviewer Confidence:**

3: Pretty sure, but there's a chance I missed something. Although I have a good feel for this area in general, I did not carefully check the paper's details, e.g., the math, experimental design, or novelty.

---

> ### Author Rebuttal · Authors · 2023-08-28
>
> Thank you for your valuable comments and feedback.
>
> > mBLIP is a minor modification. The proposed mBLIP does not always work.
>
>
> We would like to highlight that our contribution goes beyond mBLIP, and a significant emphasis is placed on the EDIS dataset which is a large-scale entity-rich retrieval dataset in a practical setting. Compared to BLIP, mBLIP improves all metrics by relatively 4-12% on the distractor set and 15-26% on the full set except R@1. We conjecture that the degradation attributed to the fact that the image-query alignment is accurate enough for matching queries with the most similar candidates. Therefore, introducing high-level headlines slightly harms the results. Note that the degradation in R@1 is much less severe in the full set and mBLIP even improves R@1 for the validation set, which proves that mBLIP is more effective than BLIP in a more practical setting.
>
> As for CLIP, we believe that the gap originates from the differences in pre-training data for BLIP and CLIP. Note that BLIP is pre-trained on COCO+VG+Conceptual3M+SBU+Conceptual12M where named entities are deliberately filtered out deliberately filtered out from the text. Though the pre-training data of CLIP is unknown, we find out that CLIP is inherently better than BLIP in understanding named entities. As a proof, we run image retrieval on the full TARA [1] development set. CLIP outperformed BLIP by a large margin. As EDIS is an entity-rich dataset, the gap is caused by the pre-training stage instead of mBLIP design or fine-tuning strategy.
> | Model       | R@1         | R@5         | R@10        |
> | ----------- | ----------- | ----------- | ----------- |
> | CLIP        | 66.42       | 87.83       | 92.27       |
> | BLIP        | 41.73       | 66.36       | 75.55       |
>
>
> > Table 4, we can find that BLIP performs better than mBLIP in two situations.
>
> For the gap in R@1, please refer to our response to the first reason to reject. The gap at R@1 in Table 4 is the same as in Table 3 while the purpose of Table 4 is to demonstrate the effectiveness of our feature-fusion design in mBLIP.
>
> > Evaluation on MSCOCO or Flickr30K
>
> Thank you for your feedback. While MSCOCO or Flickr30K are standard datasets for image-text retrieval, please note that 1) these datasets do not have the “headlines” for mBLIP to perform multi-modal fusion. The candidates are single modalities, either image (if performing text-to-image retrieval) or text (if performing image-to-text retrieval). EDIS addresses a different setting where each candidate contains multimodal information (image+headline). 2) These datasets only contain objects of general categories instead of named entities.
>
> > Questions: will you release the EDIS datasets
>
> Please be assured that we will release the full datasets, including raw images, annotation files, and the mBLIP scripts after the anonymity period.
>
> [1] Fu, Xingyu, et al. "There’s a time and place for reasoning beyond the image." ACL 2022.

---

### Meta-Review · Area_Chair_EyZp · 2023-09-09

**Recommendation:** 5

**Metareview:**

This paper brings the EDIS dataset that can evaluate the image retrieval task in news domain. This news domain challenges the model with more common sense knowledge and also fine-grained entity understanding. The reviewer raises the concern of the lack of novelty in proposing methods, which I partly agree with but the main novelty of the dataset paper will be mainly evaluated based on the data. Methodology would be considered more as a bonus.

Overall I feel that this dataset can be helpful for the community given that the authors promise on releasing dataset.

---

### Decision · Program_Chairs · 2023-10-07

**Decision:**

Accept-Main

**Comment:**

This paper brings the EDIS dataset that can evaluate the image retrieval task in news domain. This news domain challenges the model with more common sense knowledge and also fine-grained entity understanding. The reviewer raises the concern of the lack of novelty in proposing methods, which I partly agree with but the main novelty of the dataset paper will be mainly evaluated based on the data. Methodology would be considered more as a bonus.

Overall I feel that this dataset can be helpful for the community given that the authors promise on releasing dataset.